# Local Differential Privacy in Asynchronous Federated Learning

## Abstract

Federated Learning (FL) allows multiple parties to collaboratively train a machine learning (ML) model without having to disclose their training data. Clients train their own models locally and share only model updates with an aggregation server. The first FL deployments have been in synchronous settings, with all clients performing training and sharing model updates simultaneously. More recently, *Asynchronous FL* (Async-FL) has emerged as a new approach that allows clients to train at their own pace and send/receive updates when they are ready.

While FL is inherently less privacy-invasive than alternative centralized ML approaches, (aggregate) model updates can still leak sensitive information about clients' data. Therefore, FL algorithms need to satisfy Differential Privacy (DP) to provably limit leakage. Alas, previous work on Async-FL has only considered Central DP, which requires trust in the server, and thus may not always be viable. In this paper, we present the first technique that satisfies *Local DP* (LDP) in the context of the state-of-the-art aggregation algorithm for Async-FL, namely, Fed-Buff. We experimentally demonstrate on three benchmark FL datasets that our LDP technique performs equally well and, in some cases, better than FedBuff with Central DP. Finally, we study how the *staleness* of the model updates received by the asynchronous FL clients can be used to improve utility while preserving privacy under different attack setups.

## 1 Introduction

Federated Learning (FL) is an emerging machine-learning paradigm that enhances privacy in distributed learning environments. The main idea is to let clients train their own local models, on their local dataset, but collaborate to build a joint global model, by exchanging model updates through a (central) aggregator server Kairouz et al. (2021b). FL deployments are gaining traction in various settings, including language models, vision, fraud detection, etc. Hard et al. (2018); He et al. (2021); Naseri et al. (2022a).

In traditional FL instantiations, all clients simultaneously train their models and send updated models to the aggregation server. In reality, however, many applications involve large numbers of clients running from diverse sets of devices with heterogeneous resources. Thus, only a small fraction of them might be available at the same time for training and exchanging model updates – e.g., due to limited connectivity, bandwidth, battery, or computing power. This leads to conflicting objectives: on the one hand, larger numbers of clients should, in theory, yield better accuracy; on the other, increasing concurrency in FL training leads to diminishing returns in the speed of model convergence and quality of the model. Due to the heterogeneity of clients' devices and data distributions, there might be increasing numbers of clients, aka *stragglers*, taking longer to complete local training, which ultimately stretches out the time it takes to complete each round of training, hampering utilization.

To overcome these issues, Asynchronous FL (Async-FL) Wang et al. (2021) allows clients to train their models at their own pace, and send updates whenever they are ready. In other words, Async-FL does not require all clients who are participating in model training to finish their computations before proceeding to the next round. This alternative results in faster training of the global model and ultimately better scalability and improved ability to handle stragglers.

Either way, FL is potentially vulnerable to different sets of attacks that target robustness and privacy De Cristofaro (2021); Lyu et al. (2020). The former involves an adversary aiming to compromise

the accuracy and reliability of the global model. Robustness attacks can take various forms, such as data and model poisoning attacks Enthoven and Al-Ars (2021). The latter focus on compromising the confidentiality of the data and the clients' privacy, e.g., through membership inference, model inversion, or property inference attacks Melis et al. (2019); Nasr et al. (2019).

To mitigate privacy concerns, techniques have been proposed that integrate Differential Privacy (DP) guarantees McMahan et al. (2017); Geyer et al. (2017); Truex et al. (2020). DP in FL bounds information leakage by adding carefully calibrated noise to the model updates. It can take one of two forms: Local or Central DP. The former involves a trusted central server adding DP noise to the aggregate updates as received by individual clients, while the latter incorporates DP noise directly on the updates locally on each client, before sharing them with the server for aggregation. A significant limitation of central DP is its reliance on a trusted server. Additionally, it does not effectively counter attacks like membership inference, model inversion, and property inference, which exploit potential malice within the server. This underscores the critical role of Local DP in addressing such cases.

To the best of our knowledge, prior work in Async-FL has only studied Central DP Nguyen et al. (2022). As mentioned, this requires the server to be trusted with access to the clients' individual updates, and the responsibility of correctly adding DP noise to the aggregated updates. The first issue could, in theory, be mitigated through the use of secure multi-party computation (i.e., performing secure aggregation so that the server can only recover the aggregates); however, this often comes with prohibitively high computational overhead. The second issue still remains unmitigated. This prompts the need to build Async-FL techniques that satisfy Local DP (LDP), allowing clients to enjoy sound privacy guarantees vis-à-vis both the server and other adversarial clients.

In this paper, we introduce the first technique (to our knowledge) to guarantee LDP in Async-FL. We base our method on FedBuff Nguyen et al. (2022), the state-of-art Async-FL technique, but modify it to achieve LDP guarantees without compromising utility. Our experimental evaluation, conducted on both image classification and language model tasks on benchmark FL datasets, attests to the feasibility of our approach, with performance comparable to state-of-the-art buffered Async-FL with Central DP.

Finally, we present a novel approach (i.e. staleness powered LDP) that leverages the concept of staleness, i.e., the difference in timing between updates from different clients, for privacy. Our intuition is that clients can decrease the amount of noise they need to add when their updates are expected to be stale and, thus, increase the overall utility of the global model—while maintaining the same level of privacy for the system.

## 2 BACKGROUND & PRELIMINARIES

### 2.1 ASYNCHRONOUS FEDERATED LEARNING (ASYNC-FL)

In the traditional FL paradigm, clients coordinate (synchronize) their training process. More precisely, they train local models on their datasets and periodically synchronize them with a central aggregator server. This synchronization, initiated and orchestrated by the server, ensures alignment among clients and collective updates of the global model in each round. There often are two main challenges in synchronous FL: scalability and dealing with stragglers.

Scalability is a critical concern, particularly in large-scale settings with a large number of clients, as only a proportion of them may be accessible at any given time for training. Managing the training process across many clients can thus prompt significant challenges, encompassing tasks such as efficient communication, minimizing network congestion, and optimizing computational resources to accommodate the scale of the FL system. Additionally, the presence of stragglers, i.e., clients whose training progress is considerably slower than others, causes delays in the synchronization process and further complicates the learning process.

**Async-FL.** To overcome these issues, the notion of Async-FL has been proposed as a variant to the FL framework where the communication between the clients and the central server is not coordinated in real-time Wang et al. (2021). The clients are not required to wait for other clients to complete their training rounds before sending their updates to the server; rather, they can initiate communication with the server as soon as they complete their local training. This allows for more flexibility in the

training process, as clients can proceed at their own pace and contribute updates whenever they are ready.

While Async-FL addresses some challenges, it also introduces new ones, including dealing with the heterogeneity of client devices, outdated updates, communication bottlenecks, etc. Nonetheless, various techniques in Async-FL can be used to ensure an effective training process Xie et al. (2022); van Dijk et al. (2020); Chai et al. (2021); Chen et al. (2016); Wu et al. (2020); Li et al. (2019). In this paper, we focus on Buffered Async-FL, which has been shown to be more efficient than other Async-FL methods.

**Buffered Async-FL.** Nguyen et al. Nguyen et al. (2022) present a framework for Async-FL called FedBuff. Fedbuff uses a server-side buffer for aggregation incorporating client-level DP (we introduce DP in Section 2.2 below). More precisely, its implementation uses the Differentially Private Follow-the-Regularized-Leader (DP-FTRL) technique proposed in Kairouz et al. (2021a). Unlike other Async-FL techniques, the server model is not immediately updated upon receiving each client's update; rather, the updates are stored in a buffer. A server update is only executed when the buffer includes $K$ client updates. Nguyen et al. Nguyen et al. (2022) argue that $K = 10$ is the optimal buffer size; thus, we use the same value in the rest of this paper.

**Staleness.** In Async-FL, staleness describes the discrepancy between the clients' local models and the global model Dai et al. (2019). It can occur for various reasons, e.g., network delays, slow or overloaded clients, and hardware or software failures. Staleness can negatively impact model performance since updates from stale clients may not be consistent with the current state of the global model. To mitigate this issue, the server could impose a maximum delay constraint on client updates or adjust the learning rate Zhang et al. (2016); alternatively, one could assign weights based on the newness of the updates Nguyen et al. (2022); Huba et al. (2022).

## 2.2 DIFFERENTIAL PRIVACY (DP)

DP provides statistical guarantees with respect to the information an adversary can infer from the output of a randomized algorithm. In other words, it provides an unconditional upper bound on the influence of a single individual on the output by adding noise Dwork et al. (2014).

**Definition 1. Differential Privacy.** *A randomized mechanism $M$ provides $(\epsilon, \delta)$-differential privacy if, for any two neighboring databases, $D_1$ and $D_2$ that differ in only a single record, and for all possible outputs $S \subseteq Range(A)$:*

$$P[M(D_1 \in A)] \leq e^\epsilon \cdot P[M(D_2 \in A)] + \delta \tag{1}$$

The $\epsilon$ parameter (aka *privacy budget*) is a metric of privacy loss. It also affects privacy-utility trade-offs, i.e., lower $\epsilon$ values indicate higher levels of privacy, likely, at the cost of reduced utility. The $\delta$ parameter accounts for a probability on which the upper bound $\epsilon$ does not hold. The amount of noise needed to achieve DP is proportional to the *sensitivity* of the output; this measures the maximum change in the output due to the inclusion or removal of a single record.

**DP in FL.** In the context of FL, DP can be implemented in various ways, including Central or Local DP Duchi et al. (2013); Truex et al. (2020); Naseri et al. (2022b); Geyer et al. (2017); McMahan et al. (2017). In Central DP, the server is trusted to inject noise into the updates received from the clients during aggregation. By contrast, in Local DP (LDP), the clients add noise locally before sending the model updates to the server.

**LDP in FL.** LDP enables the collection of sensitive data by incorporating local perturbations, eliminating the dependency on a trusted server. $M$ is a randomized algorithm which takes a given vector $v$ as input and outputs a perturbed vector $v^*$ The concept of $\epsilon$-LDP is applied to $M$, with $\epsilon$ representing the privacy budget as following:

**Definition 2. Local Differential Privacy.** *A randomized algorithm $M$ satisfies $\epsilon$-LDP if and only if the following is true for any two possible inputs $v, v' \in V$ and output $v^*$:*

$$P[M(v) = v^*] \leq e^\epsilon \cdot P[M(v') = v^*] \tag{2}$$

## 3 LOCAL DP IN BUFFERED ASYNC-FL

In this section, we present our algorithm implementing LDP in Buffered Async-FL.

---

**Algorithm 1:** Local Differential Privacy in Buffered Async-FL (FedBuff+LDP)

---

1 **Function** `Main()`:
2     Initialize: model $\theta_0$  Buffer $B$; **while** *the model is not converged* **do**
3         $c \leftarrow$ sample available clients
4         run `CLIENT-TRAIN` (*params*) asynchronously on $c$
5         **if** *receive client update* **then**
6             $\Delta_i \leftarrow$ update from client $i$
7             $B.insert(\Delta_i)$
8         **end if**
9         **if** $B.isFull()$ **then**
10             apply_staleness_control($B$)
11             $\Delta^{-r} \leftarrow$ average_client_updates($B$)
12             $\theta^{r+1} \leftarrow \theta^r - \eta_s \Delta^{-r}$
13             $B.empty(), \Delta^{-r} \leftarrow 0, r \leftarrow r + 1$
14         **end if**
15     **end while**
16 **return** $\theta$
17 **Function** `CLIENT-TRAIN` (*clipping norm S, dataset D, sampling probability p, noise magnitude $\sigma$, client learning rate $\eta_c$, Iterations E, loss function $L(\theta(x), y)$*)**:**
18     Initialize $\theta_0$
19     **for** *each local epoch i from 1 to E* **do**
20         **for** *(x, y) $\in$ random batch from dataset D with probability p* **do**
21             $g_i = \nabla_\theta L(\theta_i; (x, y))$
22         **end for**
23         $g_{batch} = \dfrac{1}{p \cdot D}(\Sigma_{i \in batch} g_i \cdot min(1, \dfrac{S}{\|g_i\|_2}) + N(0, \sigma^2 I))$
24         $\theta_{i+1} = \theta_i - \eta_c(g_{batch})$
25     **end for**
26 **return** $\theta_c$

---

**General Form of Async-FL.** We assume the presence of $C$ distributed clients. $D_c$ denotes client $c$'s data, and $n_c$ is the number of data points on $c$, i.e., $n = |D_c|$. $\theta_c$ is the local model parameter.

The optimization function of each client is:

$$f_c(\theta_c) = \frac{1}{n_c} \sum_{i \in D_c} l_i(x_i, y_i, \theta_c) \tag{3}$$

where $l$ represents the loss function of the corresponding data point. Consequently, the aggregated optimization function is:

$$F(\theta) = \sum_{c=1}^{C} \frac{n_c}{C} f_c(\theta) \tag{4}$$

where $\theta$ is the model at the server side, and the goal is to find the optimized one ($\theta_*$) for:

$$\theta_* = argminF(w) \tag{5}$$

**FedBuff+LDP.** In Algorithm 1, we detail our approach for achieving LDP in Buffered Async-FL. The FL process starts with the server generating a model with randomized weights; this model is then distributed to all participating clients. Each client independently trains this initial model on its respective dataset.

During the training process, and to satisfy LDP, gradients are clipped to limit their magnitude, and each client adds Gaussian noise. The resulting privacy guarantees have been investigated in prior work and formally proven Kim et al. (2021); Mahawaga Arachchige et al. (2022). In Async-FL, there is no concept of rounds; each client sends its updates to the server as soon as they are available. The server aggregates these updates by inserting them into a buffer. When the buffer becomes full, the

server performs a staleness control on the updates before aggregating them to ensure that all updates are consistent with the current model. This phase follows the same approach as Nguyen et al. (2022); Huba et al. (2022) in handling stale updates – we discuss this step in more detail in Section 5. Once the check is completed, the server aggregates the updates to obtain a new model (with a new version denoted as $r$), which is then distributed to all clients for further training.

## 4 EXPERIMENTAL EVALUATION

In this section, we present an experimental evaluation, conducted on three different datasets/tasks, of our novel algorithm for LDP in Async-FL.

**Datasets and Model Architectures.** Our experiments use three different datasets: CIFAR-10 Krizhevsky (2009), CINIC-10 Darlow et al. (2018), and Reddit-comments.[1] CIFAR-10 includes 60,000 labeled images (50,000 training and 10,000 testing), each depicting one of ten object classes, with 6,000 images per class. CINIC-10 has 10 classes and 270,000 images, 180,000 of which are used for training and 90,000 for testing. To train on CIFAR-10 and CINIC-10 datasets, we use the lightweight ResNet18 CNN model He et al. (2016). We also consider a word prediction task using the Reddit-comments dataset. Following past work Bagdasaryan et al. (2020); Inan et al. (2016); McMahan et al. (2017); Press and Wolf (2017), we use a model with a two-layer Long Short-Term Memory (LSTM) and 10 million parameters trained on a chosen month (September 2019) from the Reddit-comments dataset, and we filter users with a number of posts between 350 and 500. Our training setup is similar to past work Bagdasaryan et al. (2020); Naseri et al. (2022b), and our dictionary is restricted to the 30K most frequent words (instead of 50K) to speed up training and boost model accuracy.

**Setup.** To partition the CIFAR-10 and CINIC-10 datasets, we follow the approach of past work Nguyen et al. (2022); Hsu et al. (2019) and use a Dirichlet distribution with parameter 0.1 to divide the dataset among 5000 non-iid clients. For Reddit-comments, the filtered-out users are recognized as clients with their posts as their training data. Following Nguyen et al. (2022); Huba et al. (2022), we set the buffer size to 10. We experiment with different values of concurrency, i.e., the number of clients training at a particular time. We sample the staleness of clients from a half-normal distribution with $\sigma = 0.5$.

**Results.** In Figure 1, we report the model performance with respect to test accuracy of FedBuff+LDP (our algorithm) compared to FedBuff+DP-FTRL (providing Central DP) across an increasing number of communication trips, which encompass download, computation, and upload. Then, in Figure 2, we present the accuracy results of each model for different privacy budgets, for the three datasets and models and the two tested methods.

Overall, our experiments show that, under the same privacy budget (i.e., with the same epsilon values), FedBuff+LDP provides better accuracy than FedBuff DP-FTRL. It also does this faster, i.e., FedBuff+LDP enables the construction of a global model that achieves higher accuracy faster with respect to communication trips. Note that using the same epsilon values in Local vs. Central DP variants does not necessarily imply the same level of privacy since these values have different meanings in either setting.

Finally, Table 1 shows the number of communication trips needed to reach a certain accuracy for the three datasets. The results are averaged over five random runs, and the standard deviation is computed based on these runs. Again, we observe that FedBuff+LDP picks up better accuracy and faster – i.e., in fewer trips – than FedBuff+DP-FTRL.

## 5 USING STALENESS AS PRIVACY

As mentioned in Section 1, we posit that the concept of staleness could be used to improve utility while maintaining the same level of privacy. More precisely, we set to incorporate staleness while adding noise in the Local DP setting when the adversary is one or more clients and not the server. In this section, we show how to realize this approach and show that it can indeed enhance utility without compromising privacy.

---

[1] https://bit.ly/google-reddit-comms

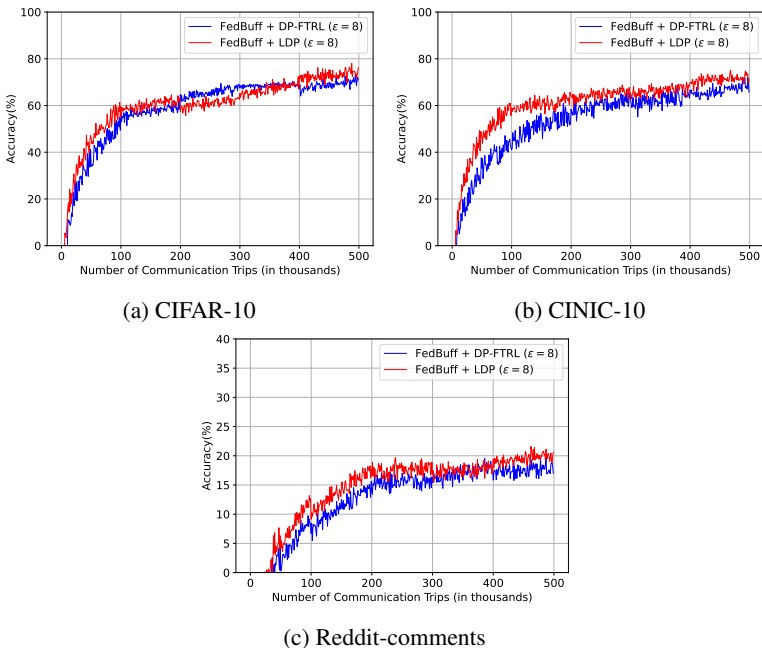

Figure 1: Main task accuracy in buffered async FL with LDP in different communication trips.

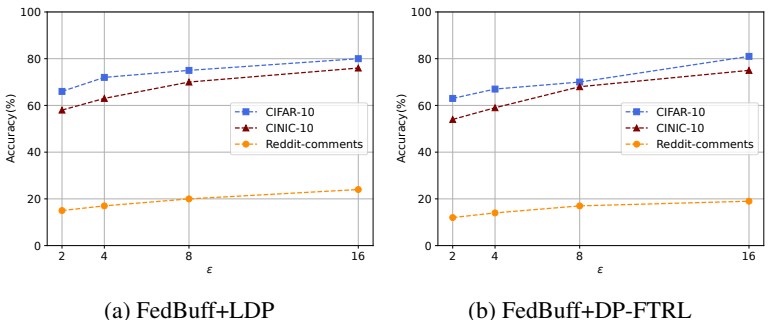

Figure 2: Main task accuracy in buffered async FL with LDP for three datasets with varying privacy budgets $\epsilon$ (lower $\epsilon$ provide better privacy).

## 5.1 STALENESS

**Definition.** Following Nguyen et al. (2022), we denote staleness as $\tau$, which is defined as the difference between the model version that a client uses to start local training and the server model version at the time when a client uploads its model update.

As performed in past work Huba et al. (2022), the staleness control phase in Algorithm 1 essentially does the following: assuming there is an update $\Delta_i$ from client $i$ with a staleness value $\tau_i$ in the buffer, the server gives the weight $w_i$ to update for

$$w_i = \frac{1}{\sqrt{1 + \tau_i}} \tag{6}$$

**Intuition.** In the rest of the section, we present an experimental evaluation demonstrating that such modified weights can compensate for lower levels of added DP noise, leading to improved utility of the global model, while providing effective protection against membership inference attacks (reviewed below).

| Dataset | Accuracy | FedBuff w/LDP | FedBuff w/DP-FTRL |
|---------|----------|---------------|-------------------|
| CIFAR-10 | 60% | $137.8 \pm 3.9$ | $195.2 \pm 4.2$ |
| CINIC-10 | 60% | $154.1 \pm 6.2$ | $233.7 \pm 5.3$ |
| Reddit-comments | 15% | $176.1 \pm 9.1$ | $214.8 \pm 7.5$ |

Table 1: Average $\pm$ standard deviation number of communication trips (in thousands) to reach a target accuracy in the three datasets with $\epsilon = 8$ and $K = 10$. Standard deviation is computed over 5 random runs of each setting with different seeds.

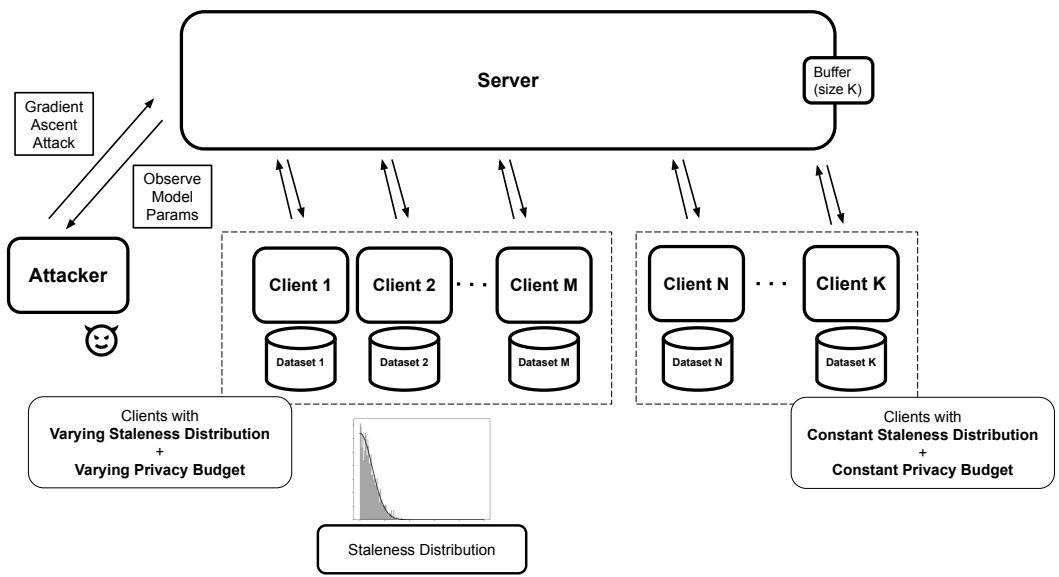

Figure 3: Adversarial Model – Active Membership Inference Attack against the Training Dataset of all $K$ Clients in Buffered Async-FL.

The main intuition when applying staleness is that updates sent from clients later (i.e., with higher staleness) will be down-weighted to affect the global model in a smaller fashion. As a result, such clients who anticipate sending their updates with higher staleness, can proactively reduce the amount of DP noise they add. This ultimately allows us to achieve the same level of privacy at the system level, while improving overall utility at the global model. Note that the level of privacy is in relation to another adversary client at the global level, and not at the personal level of each client.

## 5.2 ADVERSARIAL MODEL

**Membership Inference Attacks (MIA).** We consider an adversary attempting to mount membership inference. In machine learning, the goal of Membership Inference Attacks (MIAs) is to determine whether a specific data point was included in the training dataset Shokri et al. (2017). In the context of FL, the attacker may be the aggregation server or any of the clients. Adversaries could also be passive or active: the former only monitor parameters, while the latter also manipulate inputs to facilitate the attack.

**MIA in Async-FL.** LDP allows clients to protect their data from privacy attacks originating from both the server and other clients. In our work, we consider an attack model where one of the clients assumes the role of an attacker. More specifically, as depicted in Figure 3, we consider an adversary carrying out an active MIA Nasr et al. (2019) to determine whether a particular data point exists in any of the datasets held by clients 1 to $K$.

During training, the adversary (an active client) can intentionally manipulate the target model to extract additional information about its training set. We follow the attack introduced by Nasr et

| Dataset | Main Task Acc | Attack Acc |
|---------|---------------|------------|
| CIFAR-10 | 85.4% | 81.2% |
| CINIC-10 | 83.5% | 79.6% |

Table 2: Performance of the MIA attack with no defense.

al. Nasr et al. (2019), where the attacker performs a gradient ascent on a specific data point to infer its membership by updating the local model parameters in a way that maximizes the loss on that data point. It then sends the updates to the server, and if the target data point belongs to a client's training set, this comes with a reduction of the gradients of the loss function on that particular data point. This reduction can be identified by the inference model, enabling the distinction between members and non-members.

## 5.3 Experimental Evaluation

**Setup.** For these experiments, we focus on the CIFAR-10 and CINIC-10 datasets. We follow the same number of clients, model architecture, and buffer size as in the experiments presented earlier in Section 4. We consider a worst-case scenario attack, whereby the attacker targets *all* other clients and attempts to infer whether a specific data point exists in their respective datasets. As we aim to make the attack more effective, the attacker's updates are consistently chosen to be inserted into the buffer immediately as it responds to the server without any delay.

We assess the performance of the attack by randomly sampling 100 member and 100 non-member instances from the target model. To evaluate the performance of the attack, we measure the accuracy of both the main task and the attack. This allows us to assess the impact of the attack on the performance of the targeted task as well as the success rate of the attack itself.

Regarding staleness applied per client, we remind the reader that in the previous experiments, staleness per client (i.e., how much time it takes to respond with their update to the server) is sampled by a half-normal distribution, with standard deviation $\sigma = 0.5$. To measure the impact of staleness under this new setup, we divide the clients into two same-size groups: (1) clients 1 to $M$ and (2) clients $N$ to $K$. We vary the staleness distribution of group 1 by changing $\sigma$ of the half-normal distribution while maintaining a consistent staleness distribution for the remaining clients $N$ to $K$. By varying the staleness distribution of clients 1 to $M$, we can simulate different levels of staleness and observe its effect on the main task and attack accuracy. Keeping the staleness distribution constant for clients in group $N$ to $K$ provides a baseline or reference point for comparison. By maintaining a consistent staleness level for this group, we can isolate and analyze the specific impact of varying staleness in clients 1 to $M$.

To use this staleness-for-privacy approach, concurrently, the clients in the first group (i.e., 1 to $M$) increase the staleness of their updates and decrease the LDP noise they apply on their local models (thus increasing the privacy budget available). In practice, this is feasible as clients have the ability to keep track of the most recent model version and estimate their staleness. Also, note that clients $N$ to $K$ apply the same level of privacy budget as before.

We perform MIA using variable privacy budget (controlled by $\epsilon$) and staleness distribution (controlled by $\sigma$). We measure and compare the accuracy of the global model trained under these conditions of privacy and staleness (Main Task Acc) and the accuracy of the attacker's model (Attack Acc).

**Results.** Table 2 reports Main Task and Attack Accuracy when no DP is applied. The attack is performed by the attacker against all other clients datasets (client 1 to $K$). The experiment demonstrates that the attack is effective, surpassing a random guess baseline accuracy of 50%. Additionally, the main task accuracy decreases only slightly, indicating the minor impact of the attack on overall performance.

Then, in Table 3, we report the results of applying different privacy budgets and different scales of staleness distributions on clients 1 to $M$. As expected, decreasing the privacy budget with the same staleness scale mitigates the attack (baseline is $50\%$) better but provides worse utility.

| | | CIFAR-10 | | CINIC-10 | |
|---|---|---|---|---|---|
| $\epsilon$ | $\sigma$ | Main Task Acc | Attack Acc | Main Task Acc | Attack Acc |
| | 0.5 | 69.0% | 51.2% | 68.9% | 53.1% |
| 3 | 1 | 69.3% | 51.0% | 68.5% | 52.9% |
| | 2 | 69.2% | 50.5% | 68.4% | 50.7% |
| | 0.5 | 76.3% | 57.9% | 73.8% | 60.5% |
| 7 | 1 | 77.1% | 56.1% | 73.6% | 57.7% |
| | 2 | 71.2% | 52.5% | 73.5% | 55.1% |
| | 0.5 | 80.0% | 64.6% | 79.8% | 66.4% |
| 12 | 1 | 79.7% | 59.9% | 79.5% | 63.9% |
| | 2 | 80.3% | 57.2% | 79.4% | 59.8% |

Table 3: Main task and attack accuracy with different staleness distribution scales ($\sigma$) and privacy budgets ($\epsilon$).

Moreover, keeping the privacy budget constant while increasing the staleness results in lower attack accuracy. However, Main Task Accuracy stays around the same level. For example, in the case of CINIC-10, if we keep the privacy budget at $\epsilon = 12$ and increase the staleness scale ($\sigma$) from $0.5$ to $2$, we get a reduction in Attack Accuracy from $66.4\%$ to $59.8\%$.

Alternatively, by increasing the staleness of clients $1$ to $M$, and simultaneously raising the privacy budget, we can achieve equivalent mitigation of the attack with improved utility. For instance, in the case of CIFAR-10 with $\epsilon = 3$ and $\sigma = 1$, Main Task Accuracy is $69.3\%$ while Attack Accuracy is $51.0\%$. By increasing the staleness scale to $\sigma = 2$ and privacy budget to $\epsilon = 7$, we achieve similar Attack Accuracy($52.5\%$) while obtaining superior utility (Main Task Accuracy of $71.2\%$).

## 6 CONCLUSION

In this paper, we presented a novel algorithm integrating Local Differential Privacy (LDP) in Buffered Asynchronous Federated Learning (Async-FL). We performed an experimental evaluation on both image classification and language model tasks, over three different datasets. Overall, we compared the performance of our algorithm to the state-of-the-art Async-FL Central DP alternative and found the results to be comparable. However, our proposed LDP has the advantage that the central aggregation server does not need to be trusted.

We also explored how clients can leverage the concept of staleness in Async-FL to reduce the amount of noise added to model updates when the system is under attack by a non-trusted client. By doing this, FL clients can maintain the same level of privacy across the system while improving overall global ML utility.

**Limitations & Future Work.** While our approach presents promising advantages, when dealing with high-dimensional datasets that contain a large number of features, preserving privacy while maintaining utility becomes a challenging task in LDP. Furthermore, incorporating staleness proves to be beneficial when dealing with a client attacker. However, it remains a challenge and an item for future research to investigate how staleness can improve model utility while maintaining the same level of privacy in the presence of a fully adversarial server.

Also, there is a limit to the reduction of DP noise and the increase of staleness that an Async-FL client can introduce for the Async-FL process to work. Determining an optimal threshold for these two factors is highly dependent on the specific ML model architecture and task at hand. Identifying this threshold to achieve the desired privacy-utility trade-off is part of our future work plans.

Finally, previous studies have demonstrated the effectiveness of DP noise addition in mitigating robustness attacks in traditional FL Naseri et al. (2022b); Sun et al. (2019). In the future, we intend to investigate this aspect of FL in the context of asynchronous updates.

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
