# OpenReview forum: "Buffered Asynchronous Federated Learning with Local Differential Privacy"
_ICLR.cc/2024/Conference — Submitted to ICLR 2024_

### Official Review · Reviewer_gsJz · 2023-10-30

**Soundness:** 1 poor
**Presentation:** 3 good
**Contribution:** 1 poor
**Rating:** 3
**Confidence:** 4

**Summary:**

This paper proposes to adapt FedBuff to add local DP. After introducing this algorithm, it evaluates its performance numerically on CIFAR10 and CINIC-10, and its level of protection against membership inference attack. It gives some intuition on how the delay magnitude impact the privacy guarantees.

**Strengths:**

- the paper is short (no appendix) and easy to follow
- the idea to link staleness to different level of privacy in the context of Local Differential Privacy is interesting

**Weaknesses:**

- the results seems very limited for a top conference paper. The algorithmic contribution is straightforward (just move the addition of noise to the local part in existing algorithm), there is no mathematical contributions, and the experiments seems limited (see questions)
- the paper study the privacy with a dependence on the staleness, but it seems a bit limited as no theoretical analysis is done

**Questions:**

- Could you define properly how the privacy loss evolves according to the staleness and what hypothesis should be satisfied to have significant privacy gains ?
- In the experiments, have you done only one run or could you provide the results with intervals confidence?
- Could you compare your results to relevant baselines (FedBuff? Other asynchronous algorithms?)

---

### Official Review · Reviewer_xZb7 · 2023-10-31

**Soundness:** 3 good
**Presentation:** 2 fair
**Contribution:** 2 fair
**Rating:** 5
**Confidence:** 5

**Summary:**

Paper proposes to apply local Gaussian noise during the training process on the clients in the hope of removing the need for secure aggregation and central DP in FL setting.
While this is an interesting study (and publishing the results helps the community) I do not think the paper merits a publication in ICLR as the novelty is minimal and some of the claims are not properly explained (please refer to my comments below).
The main observation I had was that the paper is studying sample level dp (instead of user level dp) in an FL setting which is already implemented in many open sourced FL libraries. While the results are interesting the novelty is minimal also the privacy protection and the attack surfaces are different for sample level vs user level dp so the two cannot be well compared.

**Strengths:**

The paper basically studies sample level DP as a substitution for user level dp in the FL setting. Results are interesting but they apply to limited settings where there are many samples on each client. This is mostly not true for many practical applications of FL. Also the privacy protection that one gets through sample-level dp is quite weaker than the one they get with user level dp.

**Weaknesses:**

apart from the main weakness of the paper mentioned above, the other problem is that users define LDP in page 2 of the paper as a pure epsilon-dp while they apply a gaussian noise on the gradients on the client which can only align with epsilon-delta dp.

**Questions:**

have the authors consider composition on clients to calculate the final epsilon? I can see that there are noise applied in multiple steps in each client?

---

### Official Review · Reviewer_g7c5 · 2023-10-31

**Soundness:** 1 poor
**Presentation:** 2 fair
**Contribution:** 1 poor
**Rating:** 1
**Confidence:** 4

**Summary:**

Paper adds Local Differential Privacy (LDP) to the existing asynchronous FL algorithm FedBuff. Claims to be first algorithm to guarantee LDP in an asynchronous FL setting.

**Strengths:**

1. Providing privacy guarantees in asynchronous FL is an interesting area which hasn't received much coverage.
2. Clean diagrams and tables within the paper.

**Weaknesses:**

1. Paper details adding LDP into FedBuff but does not prove that its algorithm, in Algorithm 1, satisfies the LDP condition that it defines in Section 2.
    - All that is stated instead is that "privacy guarantees have been investigated in prior work and formally proven" which is not sufficient.
2. Algorithm 1 seems to simply add Gaussian noise to local gradients. This is not a novel contribution, and am a bit uncertain as to where exactly the novelty lies.
3. Related works are not covered in full detail. No discussion of previous FL works which incorporate LDP.
4. No supplementary information about experiments, proofs, or anything is provided.

**Questions:**

There are a lot of questions concerning the (lack of) theoretical results. This seems extremely problematic.

Many technical terms are unclear or not defined properly. One such example is the staleness value $\tau_i$. Is this simply just the bounded delay? This should be defined.

In Equation (6), how does this weighting change if devices are not uniformly weighted (1/n)? Also, Equation (6) seems to originate from [15], correct? This idea is also not a novelty for this work.

---

### Official Review · Reviewer_fiZQ · 2023-11-01

**Soundness:** 2 fair
**Presentation:** 2 fair
**Contribution:** 2 fair
**Rating:** 3
**Confidence:** 4

**Summary:**

The paper proposes the first local differentially private algorithm for asynchronous federated learning. It also provides privacy analysis and experimental results.

**Strengths:**

Exploiting the asynchronicity in federated learning to help improve the training efficiency as well as the privacy-utility trade-off is a timely topic that is worth studying.

**Weaknesses:**

1. The contribution of this paper is limited. The proposed algorithm is a simple combination of FedBuff and the basic mechanism of local DP. There is no technical challenge in combining these two techniques.

2. The staleness control itself is independent of the application of local DP. Contrary to the paper's claim, the staleness does not improve privacy or utility. This is because reweighting a perturbed update does not change the signal-to-noise ratio.

3. On Page 2, the paper confuses local DP with central DP.

**Questions:**

See the weaknesses.

---

### Meta-Review · Area_Chair_hDwP · 2023-12-04

**Metareview:**

The paper introduces an asynchronous local DP method for federated learning.

Strengths: the reviewers find the topic of the paper interesting as it has not been widely studied before.

Weaknesses: all reviewers find the contribution of the paper to be clearly too small for ICLR. The change to existing algorithms is small and the experiments limited. There is no theoretical analysis of the properties of the proposed algorithm or even a careful privacy proof.

**Justification For Why Not Higher Score:**

All reviewers recommend rejection (some strongly), no author rebuttal.

**Justification For Why Not Lower Score:**

N/A

---

### Decision · Program_Chairs · 2024-01-16

Reject